# Plasmon Waveguide Resonance: Principles, Applications and Historical Perspectives on Instrument Development

**DOI:** 10.3390/molecules26216442

**Published:** 2021-10-26

**Authors:** Estelle Rascol, Sandrine Villette, Etienne Harté, Isabel D. Alves

**Affiliations:** 1Université de Bordeaux, CNRS, Bordeaux INP, CBMN, UMR 5248, F-33600 Pessac, France; estelle.rascol@u-bordeaux.fr (E.R.); sandrine.villette@u-bordeaux.fr (S.V.); 2Université de Bordeaux and CNRS, LOMA, UMR 5798, F-33400 Talence, France; etienne.harte@u-bordeaux.fr

**Keywords:** plasmon waveguide resonance, lipid membrane, G-protein-coupled receptor, lipid–peptide interaction, membrane active peptide, molecular imprinted polymer, instrument development

## Abstract

Plasmon waveguide resonance (PWR) is a variant of surface plasmon resonance (SPR) that was invented about two decades ago at the University of Arizona. In addition to the characterization of the kinetics and affinity of molecular interactions, PWR possesses several advantages relative to SPR, namely, the ability to monitor both mass and structural changes. PWR allows anisotropy information to be obtained and is ideal for the investigation of molecular interactions occurring in anisotropic-oriented thin films. In this review, we will revisit main PWR applications, aiming at characterizing molecular interactions occurring (1) at lipid membranes deposited in the sensor and (2) in chemically modified sensors. Among the most widely used applications is the investigation of G-protein coupled receptor (GPCR) ligand activation and the study of the lipid environment’s impact on this process. Pioneering PWR studies on GPCRs were carried out thanks to the strong and effective collaboration between two laboratories in the University of Arizona leaded by Dr. Gordon Tollin and Dr. Victor J. Hruby. This review provides an overview of the main applications of PWR and provides a historical perspective on the development of instruments since the first prototype and continuous technological improvements to ongoing and future developments, aiming at broadening the information obtained and expanding the application portfolio.

## 1. Introduction

The first observation of surface plasmons (SPs) was made by Wood in 1902, who reported anomalies in the spectrum of light diffracted on a metallic diffraction grating [1]. Such anomalies were later proven to be associated with the excitation of electromagnetic surface waves on the surface of the diffraction grating [2]. In 1968, Otto demonstrated that the drop in the reflectivity in the attenuated total reflection (ATR) method is due to the excitation of surface plasmons [3]. In the same year, Kretschmann and Raether observed the excitation of surface plasmons in another configuration of the attenuated total reflection method [4]. These pioneering works of Otto, Kretschmann, and Raether established a basis for approaches implicating the excitation of surface plasmons and their investigation and ushered surface plasmons into modern optics. SPs are coherent electron oscillations that exist at the interface between any two materials where the real part of the dielectric function changes signs across the interface, typically a metal–dielectric interface. These are light waves that are trapped on the surface because of their interaction with the free electrons of the metal. The free electrons respond collectively by oscillating in resonance with the light wave and the created evanescent wave decreases exponentially with the distance from the surface, being particularly sensitive to events occurring at the surface.

The SPR phenomenon occurs on the surface of a metal (or another conducting materials) located at the interface of two media (usually glass and liquid) when it is illuminated by polarized light at a specific angle. This generates surface plasmons and consequently a reduction in the intensity of reflected light at a specific angle (the so-called resonance angle). The refractive index changes in close proximity to the sensor surface lead to SPR spectral changes including the resonance angle. These changes can typically be related to the increase in the concentration of biological molecules close to the surface, often referred to as mass changes. The SPR phenomenon allows for high sensitivity, direct, label-free and real-time analysis of molecular interactions in which one of the partners is immobilized on a surface. In the 1990s, there was a burst in terms of SPR sensor development in a way to characterize molecular interactions directly (without labelling) and with high sensitivity. Among those are optical waveguide resonance techniques, which have been used simultaneously to measure the refractive index and the anisotropy of biomaterials. One of these methods is dual polarization interferometry (DPI), which uses dual polarization excitation of the modes of an optical slab waveguide sensor structure with face-normal incidence end-firing of the modes. Polarization switching is achieved using a fast liquid crystal switch acting as a switchable half-wave plate and involves no mechanical realignment [5]. Although in terms of spectral acquisition, the method is in the timescale of a few ms as with plasmon waveguide resonance (PWR), the kinetic resolution could be diminished by the microfluidics associated with sample filling (flux of 40 µL/min; total cell volume 200 µL). The Reimhult group investigated in detail the kinetics of formation of a lipid membrane and demonstrated that DPI enables the real-time sensitive determination of the birefringence of the lipid bilayer, together with thickness or refractive index (with the other at a fixed value). Their study demonstrated that an understanding of the mechanistic details of an adsorption process in which conformational changes and ordering occur can be elucidated using DPI, and can be greatly enhanced through the modeling of optical birefringence. The level of information gathered via DPI in this specific application compares well with that reported for PWR studies of the same phenomena (further discussed in Section 3.1) [6]. The effect of the acyl chain structure and bilayer phase state on binding and penetration by the antimicrobial peptide HPA3 was studied using DPI, providing great insight into its mechanism of action and membrane interaction and reorganization properties [7]. The applications of DPI regarding lipid membrane interactions are numerous; a good review on this issue can be found in [8].

The coupled waveguide-surface plasmon resonance (CWSPR) method is based on a PWR chip with an additional metal layer between the waveguide layer and the buffer [9]. It combines the waveguide mode in the waveguide layer and the SPR mode at the interface between the lower metal layer and the buffer solution, leading to two TM modes. Similarly to PWR, by assessing two sets of data, disentangling the thickness and refractive index is possible. The presence of the metal layer at the outer surface provides additional options for molecular immobilization. For example, in the case of gold, self-assembled monolayers (SAM) composed of aliphatic thiols can be easily assembled as a robust and functional platform for molecular capture [10]. One disadvantage relative to other optical waveguide sensors, of which the outer layer is dielectric rather than metal, regards the decreased resistance and sensor lifetime.

Plasmon waveguide resonance (PWR) is a variant of SPR that was invented by Z. Salamon and G. Tollin in the University of Arizona (Tucson, AZ, USA) in the 1990s. Details regarding the principles of this method and the advantages relative to classic SPR are described below. The development of the method greatly benefitted from the strong collaboration with V.J. Hruby from the same university, both in terms of instrument development, as well as for the targeting of key applications, for which the method provided extremely valuable information. Instrument development and applications to specific and challenging scientific questions moved side by side, resulting in great advances, especially in the field of membrane proteins, as further discussed in Section 2 and Section 3.

### 1.1. General Principles of Plasmon Waveguide Resonance

The PWR technique falls into this category of sensors based on leaky waveguides and more specifically on the metal-clad leaky waveguide (MCLW), recently revised by Gupta [11]. As mentioned above, PWR is based on similar principles to those of SPR, with some important differences. One important difference concerns the sensor’s optical properties: while SPR uses sensors consisting of a plasmon-generating material (usually gold for its resistance in aqueous media), PWR sensors are composed of a metal layer (50 nm of silver is used because it provides sharper resonances than gold), overcoated with a thick dielectric layer (460 nm of silica) that functions as a waveguide. In PWR, plasmon and waveguide modes are coupled, and the method was first named coupled plasmon waveguide resonance (CPWR), later changed to PWR for the purpose of simplicity [12]. The presence of the waveguide has important consequences, namely, the fact that PWR allows for resonances to be obtained with both *p*-(perpendicular to the sensor surface) and *s*-(parallel to the sensor) polarized light. This is not the case for SPR, in which resonances are only obtained with *p*- polarized light.

In order to understand the physical basis for the appearance of an *s*-polarized resonance as a result of adding a dielectric layer (silica in our case) on top of the metal layer, we must revisit the principles of the SPR phenomenon. According to the electromagnetic theory, thin films are characterized by a complex dielectric constant that includes the refractive index *n* and the extinction coefficient *k* (i.e., *n* − *ik*)^13^. In the optical region of the electromagnetic spectrum this parameter is equal to the ratio of light velocity in a vacuum (*c*) to that in a medium (*v*), and is equivalent to the optical admittance (*Y*). The optical admittance is defined by the ratio of the amplitudes of the electric and the magnetic fields of the electromagnetic wave, according to the equation:*Y = C/B = c/v = n − ik*(1)

Using Maxwell’s equations, the propagation of the plane, monochromatic, linearly polarized, and homogeneous electromagnetic field can be characterized by a multilayer thin-film system with the following matrix equation [13,14]:(2)$$\left[\begin{array}{c} B\\ C \end{array}\right]=\overset{s}{\underset{j=1}{\prod }}\left[\begin{array}{cc} \cos\beta_{j} & i\left(\sin\beta_{j}\right)/y_{j}\\ iy_{j}\sin\beta_{j} & \cos\beta_{j} \end{array}\right]\left[\begin{array}{c} 1\\ y_{j+1} \end{array}\right]$$
where *s* is the number of layers deposited on the incident medium (glass prism).
*β_j_ = 2π (n_j_ − ik_j_)t_j_ cos α_j_/λ*(3)
gives the phase thickness of layer *j* at the appropriate angle of incidence (*α_j_*) and the wavelength (*λ*), and
*yj = (n – ik)_j_/cos α_j_*(4)

Equations (2)–(4) allow for the examination of the distribution of electromagnetic field amplitudes throughout the thin-film system, as well as the calculation of the transmittance, absorbance, and reflectance.

The reflectance (R) of a multilayer system is given by the following relationship involving the optical admittance:R = (*y*_0_ – Y)^2^/(*y*_0_ + Y)^2^(5)
where *y*_0_ is the admittance of the incident medium (glass prism). The incident medium must be free from absorption, so that *y*_0_ is real and equals *n*_0_ (see Equation (1)). Equation (5) describes a reflectance spectrum, i.e., reflectance as a function of the incident angle α of a beam of monochromatic light (i.e., the constant λ). This corresponds to the setup used in PWR, to recall that it is also possible to have a setup in which α is fixed and λ is varied (this is not discussed herein).

Analysis of the optical admittance shows that beyond the critical angle, the emergent wave in the final medium is evanescent and the admittance is positive imaginary for *p*-polarized light and negative imaginary for *s*-polarization. For a surface wave to be confined to the metal surface, the admittance exhibited by the adjoining medium must be positive imaginary and of a magnitude very close to that of the extinction coefficient *k* of the metal (i.e., only materials with a small value of *n* and a large value of *k* satisfy this condition). Such materials correspond to silver and gold, which are able generate a surface wave. For a metallic film, this condition is fulfilled only for *p*-polarization and for a very narrow range of α. Thus, coupling of the incident light to the surface wave results in a sharp dip in the total internal reflectance, which characterizes the resonance effect. For *s*-polarization, the admittance is always negative imaginary, and so there is normally no corresponding resonance.

However, in the PWR device, a dielectric overcoat layer (silica) is used to transform the admittance of the emergent medium, so that the admittance presented to the metal is positive imaginary for both *s*- and *p*-polarization. Depending on the properties of the different layers in the sensor, the system can result in a narrowing or a broadening of the range of angles over which the necessary coincidences are achieved, and hence a similar broadening or narrowing of the resonances. As in SPR, PWR measurements are made using the ATR method of coupling the light to the deposited thin multilayers, thereby exciting resonances that result in absorption of the incident radiation as a function of either the light incident angle (with a monochromatic light source) or light wavelength (at a constant incident angle), with a consequent dip in the reflected light intensity.

This unique specificity of PWR that allows resonances to be measured with both *p*- and *s*-polarized light makes the method sensitive to both changes in mass and anisotropy that occur as a result of molecular interactions, whereas SPR is only sensitive to mass changes. One specific and very attractive feature of PWR is related to its higher spectral resolution when compared to SPR. Indeed, PWR’s spectral half-width is about 360 mdeg and 90 mdeg for *p*- and *s*-polarization, whereas for PSR the value for *p*-pol is about 3 deg. This has important consequences in the capacity to discriminate laterally heterogeneous samples, as evidenced in a glycan-lectin interaction study performed using PWR, in which the data were compared to SPR data [15]. The importance of PWR’s spectral resolution is well illustrated in the specific applications presented in Section 3.1. PWR possesses other advantages relative to SPR; this is beyond the objectives of the current article, but a good summary can be found elsewhere [15]. As will be further explained in Section 2, this potential makes PWR an ideal method to investigate the molecular interaction of thin anisotropic films such as lipid bilayers and interacting molecules.

In practice, PWR measurements consist of the resonance excitation of electromagnetic modes of the sensor by both *p*- (transverse magnetic) and *s*- (transverse electric) polarized components of laser light (continuous wave in the visible region; He-Ne; λ = 632.8 nm and λ = 543 nm) that pass through a glass prism (BK7; sensor possessing the appropriated coats as defined above) under total internal reflection (see Figure 1 for schematics). Reflected light is measured by means of a detector (photodiode). The angle of the incident light is changed by steps of 1 mdeg in order to scan for the presence of resonances. When this is the case, there are sharp decreases in the reflectance and this constitutes a PWR spectrum. Resonances are obtained with both *p*- and *s*-polarized light, as indicated above. The *p*-resonance occurs at a lower angle in our setup and is deeper than the *s*-resonance (Figure 1 insert).

Besides these two resonances, an additional interesting signal to exploit is the total internal reflection (TIR) angle that for our sensor is visible as a small peak occurring at an angle just before the *p*-resonance. Analysis of the TIR is important because when crossing changes that occur in this parameter with those observed with *p*- and *s*-polarized resonances it provides valuable information about mass changes occurring in the bulk (far from the sensor). This is possible because the sensitivity of the *p*-pol, *s*-pol, and the TIR signal to the distance from the sensor surface is different. Indeed, although *s*-pol and *p*-pol decrease exponentially with the distance from the surface (with the *s*-pol sensitivity dropping at a considerably lower distance than *p*-pol), the TIR signal is constant all the way from the sensor surface to the bulk. This has important consequences in data analysis as the TIR signal can be indicative of nonbound (bulk) material (see Calmet [16] for further details).

A PWR spectrum is characterized by the angular minimum position of the resonance, the depth and the spectral width. All these parameters may change when molecular interactions take place in the sensor surface or in close proximity, due both to mass and/or structural changes (molecular reorganization) that occur. Thus, following such spectral changes and carefully analyzing the data allows one to obtain valuable information on molecular interactions, as presented below. Information about the affinity of molecular interactions (from the pM to the mM range), the kinetics (down to ms), as well as its impact on membrane organization (anisotropy, mass density), are obtained directly (no labeling needed) and with high sensitivity (picomole quantities of material analyzed). For a review, see [17].

### 1.2. Data Analysis and Obtained Optical Parameters

Different levels of data analysis are possible that are more or less time consuming and that can provide both qualitative and quantitative information on binding affinity between the partners, the consequences that such binding produces in terms of reorganization of the molecules and the thickness of the deposited material. One of the most routinely used approaches to obtain dissociation constants between the investigated partners consists in plotting the ligand concentration (total ligand added at each titration point) versus the resonance minimum position for either of the polarizations obtained at each particular concentration at equilibrium (with no spectral changes observed with time; this reflects the ligand/protein complex). Then, data fitting by means of a hyperbolic function that describes classical ligand binding allows for the dissociation constant for the process to be obtained (examples provided in Section 3).

To obtain further insights into the nature of the changes that accompany the molecular interactions, Salamon et al. reported on a simple procedure to disentangle the contribution of the mass and structural changes that accompany the molecular interactions [18]. This procedure has been applied to decipher the nature of peptide interactions with lipid membranes (few examples [19,20,21]). This graphical method is an important development towards a broad quantitative analysis.

One further step towards a rigorous quantitative analysis has been made by the team of Wolfgang Knoll, who developed a user friendly software (WINSPALL, version 3.02, developed by Worm J. at the Max Planck Institute for Polymer Research, Mainz, Germany; reflectivity simulation program solving Fresnel equations) based on Fresnel equations and the matrices formalism. This simulates the reflectivity of a stack of one-dimensional deposition/layers to fit the experimental spectra to theoretical ones. This approach allows one to determine the five optical parameters: the refractive indices (*n_p_*, *n_s_*) and extinction coefficients (*k_p_*, *ks*) for *p*- and *s*-pol, respectively, and the film thickness *t*. Intrinsically, the measurements made by means of PWR depend on both the thickness and the complex refractive index—along a specific orientation for anisotropic material. For example, to determine the thickness of a lipid bilayer, one has to assume the complex refractive index for both in-plane and out-of-plane orientation. These refractive index values are retrieved from the literature. Conversely, obtaining the refractive index requires one to make a hypothesis on the thickness of the deposited material such as the bilayer assembly. Thus, the user needs to be careful and critical about the optical parameters that it inputs to the different layers and also to the values obtained from the fits. The robustness of the method increases greatly when the measurements are made with more than one wavelength (in close regions of the spectrum). With the optical parameters in hand, the user can, using the appropriated equations (which can be found in [22]), characterize the deposited film and the interacting molecules in terms of thickness, mass distribution, and orientation/anisotropy. This type of analysis has been carried by the teams of Tollin and Alves teams [6,22,23,24].

To illustrate how such information is reflected in a specific example of a membrane protein that changes the conformation and orientation of a lipid membrane when bound by a ligand, refer to Figure 2. Two different possible scenarios of changes in protein conformation and orientation that result from ligand addition are presented: ligand X leads to a small decrease in thickness and a decrease in anisotropy, which could be explained by the tilting of the receptor (resulting from rotation of the trans-membrane helices); and ligand Y leads to large decrease in bilayer thickness and a decrease in anisotropy and could be explained by a lateral receptor expansion and a decrease in molecular ordering as a result of helix rearrangements.

## 2. Historical Perspective on Instrument Development

This section reports on the views of the authors, with a historical perspective on PWR instrument development, from the first prototype developed by the Tollin laboratory to the second generation developed in the Alves lab, and ongoing additional updates. Certainly, other methods have been developed in parallel by other laboratories, although this is outside of the present article’s objectives.

### 2.1. The Beginning of the Story and the First Developed Prototype

Here, we present the perspective of I. Alves, a PhD student who was enrolled in the University of Arizona, under the supervision of G. Tollin and V. J. Hruby from 1999 to 2004. This report may not be complete, as the work towards the development of the first PWR prototype had already started at that time, and may lack scientific precision, as it represents the perspective of a PhD student. S. Zdzislaw and G. Tollin first developed a homemade surface plasmon sensor [26,27]. Then, the first PWR measurements acquired using that same prototype were reported soon afterward in 1997, regarding the formation and characterization of a lipid membrane [12]. The laboratory had approached Aviv (represented at the time by Jack Aviv, the company founder), which was well known for CD instrument development, to discuss the possibility of developing and commercializing this instrument. The interactions between the laboratory of Tollin and Aviv were very fruitful and a new prototype was assembled in 2000. This instrument was rather different than the first in-lab-developed instrument as it was more user-friendly and had an interface with the user for data visualization. The creation of this instrument represented a long and intense exchange between the laboratories of Tollin, Hruby, and the Aviv company. The collaboration of the Hruby lab in this project was extremely important because it brought about new and exciting research projects with many unsolved questions that could potentially be addressed via the PWR technology. For some reason, the project of instrument development with Aviv was transferred to Proterion, which continued to develop and optimize the Aviv prototype. Proterion launched the first commercial PWR at the beginning of 2000 and sold some instruments to laboratories and pharmaceutical companies. In 2004, Proterion was acquired by Wyatt, who did not pursue with the PWR instrument project. For the moment, no other company has taken over the development of PWR as far as we know. Instruments have been developed in laboratories at the University of Arizona (Saavedra lab; [28]) and in the Alves laboratory, as described below.

### 2.2. The Second Generation

The second-generation PWR, described in this section, corresponds to the instrument that was developed and implemented by Harté and Alves [6] in the Alves laboratory in 2011. The overall evolution of the PWR technique was aimed at increasing its accuracy and temporal kinetics, while having fully automated acquisition and data exportation processes. The original instrument, built by the company Proterion and based on the first homemade prototype developed by Salamon and Tollin at the end of the 1990s (US patent 5,991,488) implicated many mobile optical elements, such as mirrors and polarizers that were used to select the polarization of the incident beam and its wavelength. Each modification in the setup resulted in a delay of at least 10 s, with reliability issues and position uncertainties.

In the second generation, the decision was made to suppress all mobile parts. Moreover, to increase the temporal resolution of the system, the polarization angle of the incident light beam was adjusted to possess both electric fields, as demonstrated by E. Harte et al. [29]. A value of 45° would ensure that the incident light photons would be equally distributed between *p*-and *s*-pol, but we used an angle of 51° to favor the *s*-pol relative to *p*-pol and to improve the S/N ratio of the *s*-pol resonance, which is inherently less deep. This configuration has several advantages: the mobile polarizers became unnecessary; thus, accuracy was improved because the error in the positioning disappeared; acquisitions can be repeated without any latency and the acquisition rate is only limited by the duration of the angular scan (e.g., 1 s). In this configuration, we are able to record the total PWR signal in less than 10 s.

One additional feature of this new generation regards data acquisition and real-time analysis. In the previous software version included in the instrument and developed by Aviv and Proterion, only the acquired spectra were visualized by the user during the experiment. That means that, in order to directly analyze the experiment, the user needed to estimate or measure for each spectra the resonance minimum position and spectral depth for both polarizations as well as to monitor for any changes in TIR. This was quite time-consuming and unpractical. Therefore, with this in mind, a new program written in Labview that allows to parameterize, control, and monitor the experiment. The data are analyzed in real time, meaning that the position of the TIR and the resonance minimum angle position in *p*- and *s*- resonances are extracted on the go. Moreover, the spectral depth is also monitored. Kinetics data are obtained and the correlation between the different parameters can also be directly visualized in the form of graphs to inform users respectively on the equilibrium, the orientation/anisotropy (ratios of *p* and *s* resonance shifts) and the homogeneity of the system (which can lead to changes in the spectral depth or even the appearance of additional fringes within the same resonance, (as reported in Section 2 [16,24,30,31]). This is of particular importance during multi-step experiments such as the monitoring of the ligand binding and induced receptor conformational changes, in which it is essential to ensure that the system is at equilibrium at each point in the ligand titration in order to determine a valid dissociation constant.

The automation and reliability enable the monitoring of both fast processes, as the ligand-induced receptor conformational changes (in order of sec to min), as well as slow processes such as amyloid protein aggregation on a lipid membrane, which can last for over 24 h. Such experiments were not possible using the previous setup.

An additional feature of the current instrument setup is that it allows for multi-wavelength acquisitions, while removing the afferent mobile mechanical parts. We chose to focus on the angular sector of the rotation stage, which has a high speed (over 360°/s) and high accuracy (better than 1 mdeg). Indeed, as an angular scan for the two resonances at the same wavelength requires less than 90°, up to four angular sectors can be dedicated to up to four different wavelengths. The program is designed for such use. The wavelength can be selected (e.g., green (543 nm) or red (633 nm) He-Ne emissions were used in the laboratory, although other wavelengths can be implemented). Switching between wavelengths requires less than 1 s, since it is performed through a simple stage rotation; thus, acquisition can alternate successively between wavelengths to acquire multi-wavelength spectra in kinetic mode. With the idea of increasing the spectral acquisition speed, a specific set-up was developed in order to record four PWR signals almost simultaneously without manual positioning of the optical element. In Figure 3 we present one implementation of this setup, at four wavelengths, in which the *s*- and *p*- polarizations at each excitation wavelength were constantly directed at the sensor from four fixed directions, the sensor itself being a 90° prism placed on an extremely precise (below 10 mdeg in the positioning of the prism on its holder) rotation plate. The software automatically managed the movement of the rotating stage to position correctly the prism relatively to the incident light. The total acquisition time of the four resonances could then be performed in less than a minute, allowing for fast monitoring of the phenomenon.

### 2.3. Ongoing and Future Developments

Several technical developments have emerged from our laboratory and collaborators based on PWR technology with the idea of developing a particular feature or coupling it with another method. Often this instrument development has been triggered by the need to obtain additional information on a particular system of molecular interaction. Such instrument developments are briefly presented in this section.

#### 2.3.1. PWR Imaging

One of these developments is PWR imaging (PWRi), the development of which was motivated by the lack of information on lateral sample homogeneity, for the eventual detection and monitoring of lateral inhomogeneities and domains. The PWR setup, as described by Salamon and collaborators in 1997 [12], allows one to measure the mass and anisotropy changes of nanometer-thin 1D samples. The technique measures the reflectivity of a 9 mm^2^ sample with a probed area of the incident beam (~1 mm^2^). Samples are thought to be spatially homogeneous. Sample heterogeneity in the plane (which is laterally relative to the sensor surface) can arise either because the deposited sample is heterogeneous in nature (e.g., lipid membranes with specific lipid composition that spontaneously form lipid domains or rafts) or because its deposition has been made in a non-homogeneous manner (e.g., the lipid membrane formed has defaults or areas of the sensor are not covered) or because a certain interaction leads to heterogeneity as a result of molecular interactions and lateral reorganization (e.g., interacting membrane active peptides with lipids do not uniformly cover the surface but are deposited in domains or aggregates in certain membrane zones). With PWR there are some ways by which we could interrogate sample lateral heterogeneity, range from the simpler methods to the most sophisticated: (1) to calculate the second derivative of the angular spectra to resolve the position of the underlying resonances as applied to the characterization of amyloid aggregation on lipid bilayers [6]; (2) through the design of a new optical sensor presenting narrower resonances so that the contribution of different areas in the sensor will be resolved in properly separated resonances. This strategy was applied by Salamon to detect lateral lipid segregation in proteolipid bilayers [31,32]; (3) moving PWR into an imaging technique allow to circumvent this limitation at the price of a higher complexity and bigger data processing.

Disentangling the contribution of inhomogeneous areas which are merged into one single spectrum is not ideal if the sample is known to be heterogeneous. By imaging with a microscope the inner surface of a slightly modified PWR sensor, one can obtain images of refractive indexes and the anisotropy of a thin sample (the principle of a PWRi instrument). A proof of concept was developed and patented by Harte on synthetic materials [33] (French Patent 3,054,320). The system is ideal to monitor, for example, thin (1 to 100 nm range) samples with a patterned or non-uniform thickness and anisotropy, such as an azobenzene thin film patterned with an actinide light. Unfortunately, the limitation in terms of lateral resolution (around only a few µm) of the PWRi has greatly restricted its use for certain applications such as those related to lipid membrane domains and protein partitions into those that are central to our laboratory. This problem was solved by Elezgaray and collaborators [34] by modeling and building a plasmonic microscope presenting sub-diffraction resolution imaging of the topography and of the refractive index. This method has been applied to the study of cellular systems [35]. One drawback is the relative complexity of this setup.

#### 2.3.2. PWR with a Diverging Beam

As opposed to this sophisticated setup, Isaacs et al. [36] presented a new compact and mobile setup, characterizing the anisotropy and thickness of a homogeneous sample. The complete removal of motor or mobile opto-mechanics is achieved using a diverging beam. The angle of incidence is converted into spatial information coded in one dimension of a 2D light sensor (camera). Acquisition at the camera rate is therefore possible. This tool enables the study of anisotropic nano-layers at a low cost and video rate as the formation of a membrane lipid bilayer from vesicles.

#### 2.3.3. PWR at Multiple Wavelengths in the Visible or the IR Region

Although plasmon resonance is more commonly performed in the visible region, some laboratories have also explored the IR (near- and mid-range) spectral regions (in a non-exhaustive manner [37,38,39,40,41,42]). One of the advantages in proceeding from the visible to the IR region is that the created evanescent wave will propagate deeper into the sample, allowing the method to be applied to thicker samples such as whole cells, as reported by the Aeroti lab [38]. The other advantage is related to its capacity to discriminate between the contributions of different biological elements, as illustrated by Limaj [43]. Considering the literature on the subject, the possibility of applying PWR in the IR region to discriminate between the spectral contributions of different elements should be possible.

One limitation of PWR is related to the distinction of the relative contributions of each molecular partner in a complex system. For example, in the case of the interaction of a membrane active peptide with a membrane, it is important to separate the positive signal that reflects the total mass gain due to peptide interaction and insertion in the membrane from a potential negative signal, which could be related to a detergent effect of the peptide, and thus a reduction in the bilayer mass. In addition, in the case of ligand-induced receptor activation (as will be described in Section 3.1.1), we cannot determine from one single-wavelength PWR signal if it reflects conformational changes of the receptor alone or if there is also a lipid reorganization of the supporting membrane [16].

To circumvent this limitation, it is useful to simultaneously monitor the PWR signal at two different wavelengths, where the relative absorptions of the partners are very different. In that case, spectral analysis of the data gives one access to a full set of information, allowing one to distinguish between the contributions of both partners [22]. It makes it possible to follow structural changes, monitored by means of optical dichroïsm, occurring in either lipid or protein molecules, and thus to distinguish between changes occurring in these two phases. We can also refer to molecular polarizability and molecular shape within ordered thin films [44]. Indeed, Tollin and collaborators have applied this protocol, using 1–10 mol% of an acyl chain chromophore-labeled PC incorporated into a solid-supported PC bilayer deposited onto a hydrated silica surface. PWR measurements were made at two exciting light wavelengths, one of which was in the absorbance spectral region of the chromophore. These results were used to calculate longitudinal and transverse molecular polarizabilities, the orientation order parameter and average angle between the longitudinal axis of the lipid molecule and the membrane normal, and the molecular shape factors of the lipid molecules.

Although using dual-wavelength PWR in the visible range permits us to benefit from this strong advantage of discriminating between signals from two components of the membrane, it makes it necessary to use one chromophore-labeled partner. Shifting the PWR exciting light wavelength into the mid-infrared region, where lipids and proteins have well defined absorption bands, makes it possible to benefit from the intrinsic absorption of the partners and eliminate any potential influence of the labeling process.

We recently started to explore this expansion of PWR measurements in the infrared region. The key challenge is to adapt the optical sensor to this energy range (800–4000 cm^−1^, i.e., 4.5–12 µm), where optical indexes are not as well-known as in the visible range. Moreover, specific IR transparent materials have to be used, and the deposition and characterization of such coatings is not an easy task.

#### 2.3.4. Coupling of the PWR with Electrochemistry

Another PWR limitation concerns the lack of information regarding lipid membrane electrochemical properties, which would be extremely informative when monitoring pore formation by means of membrane active peptides as antimicrobials or in the opening of a membrane channel.

Combining the functions of electrical impedance spectroscopy (EIS) and optical characterization on a common surface is desirable. Sugihara et al. developed such a dual sensor by adding a specifically modified ITO onto an optical waveguide spectroscopy setup [45]. With this setup, they were able to monitor melittin-induced pore formation in a lipid membrane. The inclusion of an electrode (ITO) in PWR sensor design has already been described by Tollin et al. but induced a relatively poor PWR signal [46]. We have recently been working on the design of such a sensor, in which the design and simulations are of prime importance, to design a sensor that is both sensitive enough for good PWR signal, i.e., with a deep and narrow resonance, while including a thick enough conductive layer necessary for impedance coupling. Several materials are under scrutiny for this application, such as ITO, which is very interesting due to its good conductance, while being transparent in the visible range, or TiO_2_, which could also be interesting but is not widely used.

However, our first tests with ITO were not conclusive because the absorption of ITO was too strong.

## 3. Applications of the Method for the Study of Molecular Interactions

PWR has emerged as a powerful SPR variant for the characterization of molecular interactions, especially those occurring in thin films. Among the applications, a large part of them concern lipid membranes and intrinsically present (membrane proteins) or interacting molecules (e.g., membrane active peptides). For such studies, as will be described below, the sensor does not need to be chemically modified. However, other applications in which the sensor has been modified have been reported as well and will be presented below.

### 3.1. Molecular Interactions Occurring at Lipid Membranes

PWR, due to the capacity of the instrument to obtain both mass and anisotropy information, is ideal for the study of thin anisotropic films such as lipid membranes. Indeed, the first and most widely reported PWR studies were performed on lipid membranes and inserted or interacting molecules.

The first PWR publication by Tollin and collaborators reported on the formation of a lipid bilayer (black lipid membrane) using the Montal–Mueller approach and the characterization of this membrane in terms of membrane thickness and refractive indices for the two polarizations [12]. The membrane thickness and lipid organization (anisotropy) values correlated well with those reported in the literature for the supported membranes formed and characterized using other approaches, validating the PWR approach for the study of this type of films [12] (Table 1). Later, using the second-generation sensor, Alves and collaborators characterized the formation of an anionic and zwitterionic membranes in terms of the kinetics of their formation and anisotropy. This was one of the few studies reporting the real-time observation and characterization of the lipid membrane formation process [6].

Later, other studies by Alves and collaborators have reported on the formation of lipid membranes by means of another approach, the deposition of small unilamellar vesicles (SUVs) on the silica surface that form a planar lipid membrane by spontaneously bursting [16], an approach that is often used in many laboratories for diverse applications and analysis via microscopy approaches, among others. However, a more complex lipid membrane composed of unsaturated phosphatidylcholine (PC)/sphingomyelin (SM) binary mixtures, which is known to form lipid domains, was investigated through PWR using a more complex sensor design, allowing researchers to obtain sharper resonances (details can be found in [24]). Spectral simulation of the resonance curves demonstrated an increase in bilayer thickness, long-range order, and molecular packing density in going from dioleoyl phosphatidylcholine (DOPC) to palmitoyl phosphatidylcholine (POPC) to SM in the case of single-component bilayers, as expected based on the decreasing level of unsaturation in the fatty acyl chains. DOPC/SM and POPC/SM binary mixtures yielded PWR spectra that could be ascribed to the superposition of two resonances, corresponding to microdomains (rafts) consisting of PC- and SM-rich phases coexisting within a single bilayer. These were formed spontaneously over time as a consequence of lateral phase separation [24]. This lateral heterogeneous membrane was used to follow the differential insertion of membrane proteins within the two domains (described below, Section 3.1.1) [31]. More generally, model membranes formed by means of the described approaches have been used as a support for the incorporation of membrane proteins and the study of ligand interactions, as well as providing a system to characterize the affinity and kinetics of membrane active peptides, as described in the following two sections.

Another type of lipid membrane that has been employed in PWR studies concerns cell membrane fragments, which have dual interest for our applications—the fact that they present the more physiological membrane model; they harbor sugars important for certain types of studies [47]; and they can be enriched in specific proteins when those proteins are overexpressed in the cell line used to obtain the cell membrane fragments. Two different protocols have been used to immobilize cell membrane fragments on the outer silica layer of the sensor—(1) following the isolation of the membrane fraction from cells (expressing or not expressing the protein of interest) they are incubated with the sensor in the presence of 5 mM CaCl_2_ (this protocol was successfully applied to membrane fragments from mammalian cells in culture and from animal tissue [48]); and (2) the immobilization of cell fragments on the sensor pre-coated with polylysine (PLL) via the electrostatic interaction between sugars in cell membranes and PLL (adapted from Vogel et al. [49]). This allows for cell membrane fragment immobilization from cells in culture (detailed protocols found in [50,51]). This type of membranes has been used for the two applications reported below.

#### 3.1.1. Including Membrane Proteins

When investigating membrane proteins, it is essential to ensure that their physiological environment is respected, namely, that they are surrounded by lipids, detergents, or other molecules that can mimic the hydrophobic lipid environment. Moreover, lipids have also been shown to work as cofactors, being essential to maintain protein functionality [52,53]. Much of the reported work on sensors such as SPR on membrane proteins have been performed in detergent-solubilized membrane proteins, proteoliposomes, or nanodiscs with covalent attachment via modified lipids, tags on the protein, or other methods. Although such systems are very compatible with SPR and other sensor technologies, for PWR to fully take advantage of the anisotropy information that the method can provide, it is essential to maintain the orientation of the proteolipid system. Thus, PWR studies on membrane proteins have been performed in two types of systems: (1) reconstituted model membranes, in which the lipid membrane environment can be totally controlled and the quantity of inserted protein can be monitored and controlled; (2) cell membrane fragments that are immobilized on the sensor surface via PLL or in the presence of CaCl_2_. Both procedures, besides the fact that they allow cell membrane fragments to be immobilized in a rather oriented manner (with the lipid main axis perpendicular to the sensor surface and extracellular protein surfaces that can be either oriented towards the sensor surface, the bulk, or a mixture of both), they possess the advantage of being easily removed from the sensor surface by detergent washing, allowing the sensor to be reused several times.

Using the two approaches described above, different aspects of receptor activity have been investigated: (1) receptor reconstitution has been directly followed and the kinetics has been obtained for the process [16]; (2) ligand-induced receptor activation and subsequent conformational changes [16,51,54,55,56,57,58]; (3) the interaction and affinity of effector proteins to ligand-receptor complex (R/L) [29,44,59]; (4) the partitioning of the receptor into different lipid domains has been monitored and quantified [31]. Such studies, mostly implicating G-protein-coupled receptors (GPCRs), have contributed greatly to the understanding of the mode of function of such receptors (these are schematized in Figure 4; some are briefly mentioned below).

A report on the human delta opioid receptor (hDOR) revealed for the first time that there is no necessary correlation between the affinity of a G-protein to bind to the R/L complex and its capacity to undergo GTP/GDP exchange [29]. Additionally, the affinity of different G-protein subunits to the receptor occupied by different ligands revealed the high specificity that exists, with certain ligands potentiating the interaction with certain G-protein subtypes. This was one of the first pieces of evidence of a source of biased signaling in GPCRs [59].

Regarding ligand-induced receptor conformational changes, a study on the neurokinin 1 (NK1) receptor revealed the presence of two receptor isoforms, the nature of which was then further established through native mass spectrometry approaches [58].

Due to ability of PWR to be used on reconstituted systems, studies on rhodopsin have validated the fact that the lipid membrane curvature affects the Meta I-Meta II transition in rhodopsin, with the pKa for the acid–base equilibrium between the two species that changes from 6.4 to 7.3 when phosphatidylethanolamine (PE) is incorporated in the membrane [44]. In this same report, it was shown that the presence of PE in the membrane also strongly impacts transducin interaction (the G-protein specific to rhodopsin), revealing for the first time that the lipid’s impact extends further down the receptor signaling pathway. A study on the chemokine CCR5 receptor revealed that cholesterol impacts ligand affinity to the receptor, establishing a correlation with biological studies that indicate cholesterol depletion in cells to greatly decrease HIV infection (CCR5 is a co-receptor for HIV infection) [16,60].

Through the use of a sensor with a more complex array of coatings (as mentioned in Section 3.1), PWR was able to distinguish between different lipid domains in heterogeneous membranes [24]. The partitioning of hDOR between ordered and disordered membrane domains was followed and quantified by PWR in a study that demonstrated that the occupancy state of the receptor highly impacted the receptor partitioning within the two lipid domain types [31].

#### 3.1.2. Implicating Membrane Active Peptides and Small Molecule Interaction with Lipid Membranes

PWR has been used to monitor and characterize the interaction of molecules of different natures (small regions of membrane proteins, membrane active peptides, small drugs) that in their mode of action are prone to interact with lipid membranes [21,61,62,63]. In such studies, model lipid membranes of varied composition, as well as cell membrane fragments, have been used. Herein, two different research projects will be highlighted regarding (1) the mechanism of action of cell-penetrating peptides; (2) membrane interaction and domain formation induced by amyloids.

A pioneer study from Tollin and collaborators on penetratin (one of the first discovered cell-penetrating peptides (CPPs)) demonstrated that this peptide interacts very differently with zwitterionic versus anionic membranes with a biphasic event as a function of concentration observed in the second case. For both membranes, the impedance spectroscopy measurements demonstrated that the electrical resistance was not altered by peptide incorporation, whereas a decrease in membrane capacitance occurred with the same concentration dependence. The results suggest that penetratin binds electrostatically within the headgroup region of the bilayer and influences the headgroup conformation, the amount of bound water, and the lipid-packing density, without perturbing the hydrocarbon core of the bilayer (no pore formation) [46]. Later, studies by Alves and collaborators on the interaction of penetratin and other CPPs as arginine-rich peptides with model and cell fragment membranes have reinforced the idea that electrostatic interactions are important for the recognition and accumulation of CPPs at the membrane surface [19,47,64] but that this accumulation and high-affinity binding at the cell-surface do not reflect the internalization efficacy of the peptide. Thus, while this stage must be necessary to trigger membrane crossing and ultimate peptide internalization, the fact that a CPP (at least for the CPPs studied in this report) binds with stronger affinity does not necessarily mean that it internalizes more [57]. Moreover, studies performed on arginine-rich CPPs possessing a variable number of Trp residues revealed a direct correlation between the number of Trp residues and the reversibility of the interaction following membrane washing [65]. Studies on whole homeoproteins, rather than cell-penetrating peptides, by Joliot’s team have evidenced some sort of specificity in terms of the lipid headgroup, as much higher affinities were observed in the presence of membranes containing phosphatidyl inositol biphosphate (PIP_2_) when compared with phosphatidylserine. This affinity was further enhanced when both PIP_2_ and cholesterol were present, and contrarily to the mechanism of action of CPPs alone, for homeoproteins the binding affinity and translocation ability were revealed to be highly correlated [66]. Moreover, with the full characterization of peptide–lipid interaction provided by PWR in terms of the kinetics and affinity of the interaction, information about lateral distribution can also be obtained. Indeed, due to the fact that PWR resonances are quite sharp (with a high spectral resolution when compared to other plasmon resonance methods), when interacting molecules are distributed unevenly across the membrane (domain formation), the spectra can present several fringes, reflecting this heterogeneity. Therefore, the PWR method is capable of determine if an interacting molecule is distributed evenly across the lipid membrane surface or if it is distributed in a heterogeneous manner, resulting in the formation of domains (Figure 5). To be observed by PWR at the wavelengths we used, heterogeneities must be larger than about 100 nm. This PWR feature is quite valuable for the study of amyloid-type peptides with a great tendency to oligomerize, as described below.

PWR provided valuable information in the understanding of the mechanism of action of amyloid peptides in projects led by Lecomte and collaborators, who developed a model of amyloid toxicity in the yeast *Saccharomyces cerevisiae* based on toxic mutants of the HET-s prion of *Podospora* anserine [67]. The strong interest of this project was on the relation between the morphology and structure of amyloid peptides, and the toxicity expressed by their deleterious effects on biological membranes. A reported highlight was that toxic species (deleterious for the membranes) were small objects (oligomers), structured in anti-parallel β-sheets, whereas fibers structured in the parallel β-sheet had no disruptive effect on the membrane models [67]. In this study, PWR made it possible not only to establish the kinetic interaction constants of the HET-s and mutant peptides with different model membranes, but above all to highlight the presence of different forms or sizes of aggregates interacting with these membranes [6]. The addition of the toxic peptide to the lipid membrane leads to the appearance of additional resonances in the *s*-pol, which was attributed to heterogeneity in peptide assembly with the presence of regions (domains) possessing different masses and organizations. Since amyloid peptides have a great tendency to oligomerize and to form fibers, the multiple resonances seen can be attributed to the different oligomeric states of the peptides. Strong lipid reorganization has also been observed upon incubation of the oligomeric and more toxic species (oligomers) of Aβ_1-42_ (involved in Alzheimer’s disease) with membranes, whereas this was not the case of the least toxic species (monomers). Indeed, oligomeric species lead to a two-stage process associated with two different kinetic constants, starting with the anchorage of the oligomers on the membrane and followed by the induced reorganization of the lipids [30].

### 3.2. Occurring in Modified Sensor Surfaces

Apart from lipid bilayers, various layers have been deposited on optical sensors for the highly selective and sensitive detection of biomolecules using PWR. Two main examples are described here.

#### 3.2.1. Molecular Imprinted Polymers

MIPs are polymers synthesized to mime receptors, recognizing their ligands selectively. Molecular recognition elements (MRE) are artificially reproduced, integrating free chemical groups that are able to form specific non-covalent interactions with the ligand. MIPs allow for the recognition of small molecules, with a selectivity similar to natural biological targets such as receptors but developed to present a prolonged lifetime [68]. Monomers are placed in solution with the ligand of interest, which plays the role of a template for the MRE. Polymerization is then obtained by adding a cross-linker and a catalyst; then, the template is removed through intensive washings. MIPs are generally coupled to optical sensors due to their very high sensitivity [69]. Several MIPs have been developed using surface plasmon resonance (SPR) because it is the most developed technology for biosensing. Several procedures have been developed to prepare thin polymeric films imprinted with small molecules as templates. For example, an MIP was prepared on a gold surface to detect dichlorobenzidine, presenting a molecular weight of 253 g/mol [70]. Very low concentrations of this small molecule have been detected, in the sub-nanomolar range. The selectivity for the template molecule has been tested, demonstrating no difference between the imprinted polymer and the non-imprinted one for the binding of several analogues of dichlorobenzidine. MIPs have also been successfully coupled to PWR in order to selectively bind a ligand of the δ-opioid GPCR [71]. Ligand binding affinities were found in the picomolar range against the nanomolar range for the natural δ-opioid receptor, with very good reproducibility. Furthermore, the selectivity of the imprinted polymer for the template ligand has been investigated, demonstrating that other δ-opioid receptor ligands have lower affinity for the MIP than the template. As validated by this study, PWR is very interesting in this application because the selectivity and sensitivity of MIPs can easily be compared to those of natural membrane receptors such as GPCRs.

#### 3.2.2. Chemically-Modified Sensors

PWR development has been associated with various chemical functionalizations of the sensor. To compare it again with SPR, gold functionalization has been well developed by using self-assembled monolayers (SAMs), although some sensors are also covered with a silica layer. For PWR, the upper layer of the sensor is a dielectric waveguide layer such as silica, titanium dioxide, or ITO [72]. Silica functionalization has been extensively studied, leading to a large choice of advanced sensor surfaces, as shown in the following examples. First of all, the silicon waveguide can be prepared using the Stöber solution growth method, leading to a mesoporous silica thin film [73]. This sensor was able to detect changes in the refractive index, as shown by the measurement of the incident angle with different NaCl concentrations. This sensor demonstrated the ability to distinguish molecules that were able to adsorb to the surface, inner, or outer pores, and molecules were also detected in the bulk area. Finally, the sensitivity was lower but the limit of detection was better than for SPR, even for small molecules. Another example was the immobilization of gold nanoparticles (AuNPs) of 20 or 100 nm in diameter [74]. This was performed through the incubation of bifunctional siloxane 3-aminopropyltrimethylsiloxane (APTMS) on the previously deposited silica layer. This chemical group forms covalent siloxane bonds with the surface and between lateral groups and presents a primary amine that is able to form covalent or non-covalent interactions. AuNPs were immobilized through their interaction with this terminal amino group. These modified sensors, presenting two different particle densities depending on AuNPs size, were used to investigate the interaction of AuNPs with proteins. As for the previous example, PWR was able to decouple the bulk and surface effects. Another application of silica-based waveguides has been described to investigate mannose-lectin interactions [15]. In this study, the silica layer was first incubated with APTMS, which was followed by two reaction steps. Two different PWR mannose-modified sensors were prepared: one comprising a short linker group between APTMS and mannose, the other comprising a polyethylene glycol (PEG) spacer group. The selectivity of the lectin binding was investigated by comparing two different lectins. PEG spacing constituted a better functionalization strategy, enhancing the limits of detection for specific lectin interactions.

## 4. Conclusions

This review summarizes the great potential of PWR for applications on anisotropic thin films for the characterization of molecular interactions on sensor surfaces. Although most PWR applications have focused on interactions occurring in non-modified sensor surfaces, where lipid membranes are directly formed, there is also potential for studies on chemically modified surfaces, with few examples provided in this review. As this method was devised and developed in laboratories, this article provides a historical perspective on the long road of instrument development and the associated ups and downs, in which the strong collaboration of researchers with complementary expertise has been essential for the successful outcome.

## Figures and Tables

**Figure 1 molecules-26-06442-f001:**
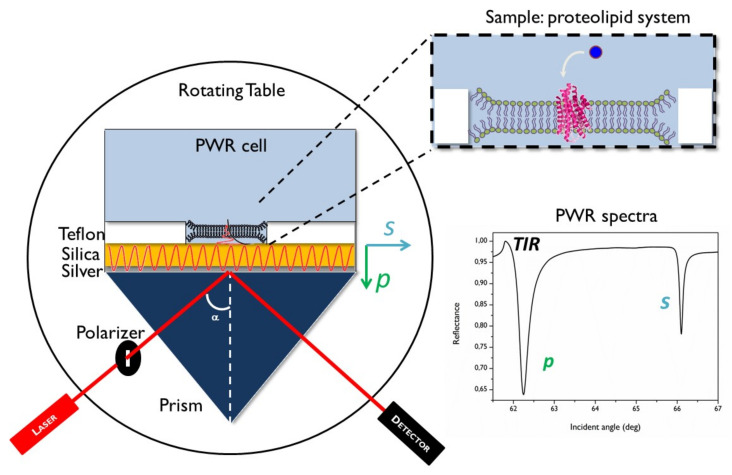
PWR setup. Optical and mechanical components (left)—the incident polarized light beam (a continuous He−Ne laser at 632.8 or 545 nm) and the rotating table allow steps of 1 mdeg. The sensor consists of a right-angle BK7 prism coated with a 50 nm layer of silver, overcoated by 460 nm of silica. Polarized light *s*- (parallel) and *p*- (perpendicular) are defined relative to the sensor surface. A proteolipid system is depicted as an example of a sample (insert top right). On the right bottom is a typical PWR spectra showing the total internal reflection angle (TIR) and *p*- and *s*-resonances.

**Figure 2 molecules-26-06442-f002:**
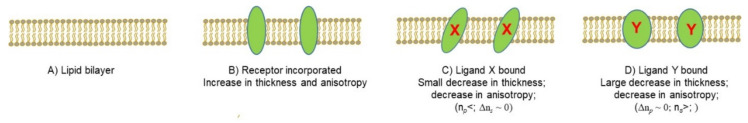
Changes in the optical properties (thickness, refractive indices, and anisotropy) of a lipid bilayer (**A**) following receptor incorporation (**B**) and ligand-induced conformational changes by two different ligands X (**C**) and Y (**D**). Adapted from [25].

**Figure 3 molecules-26-06442-f003:**
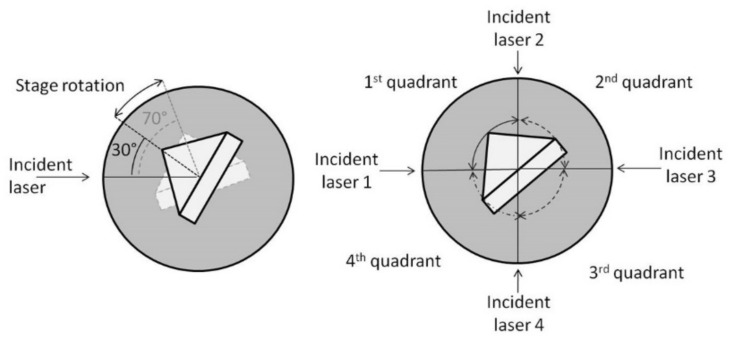
Schematics of the PWR sensor relative to the incident light. On the left we present the maximal range of the prism positioning during a typical PWR spectrum acquisition. On the right is one possible implementation of this setup using up to 4 wavelengths.

**Figure 4 molecules-26-06442-f004:**
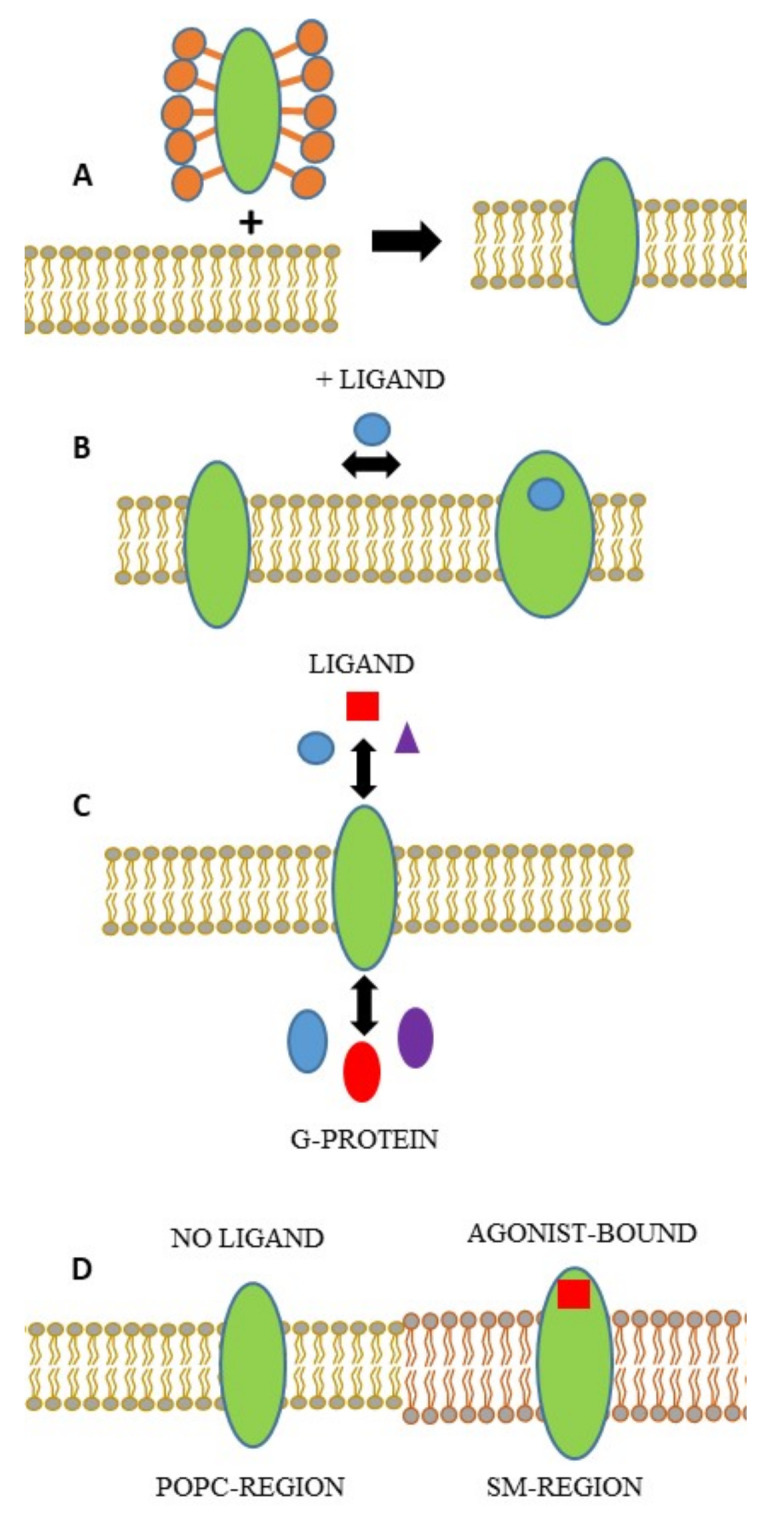
Representation of the different aspects of receptor activity that can be investigated by means of PWR. (**A**) Receptor reconstitution from detergent micelles into a lipid membrane. (**B**) Ligand-induced conformational changes of the receptor. (**C**) Ligand and effector interaction (G-protein in this case) with the receptor. The fact that different ligands can lead to the recruitment of specific G-protein subtypes is illustrated as represented by the color code. (**D**) Receptor distribution in different lipid domains, enriched in phosphatidylcholine (POPC) or sphingomyelin (SM) as a function of receptor occupancy.

**Figure 5 molecules-26-06442-f005:**
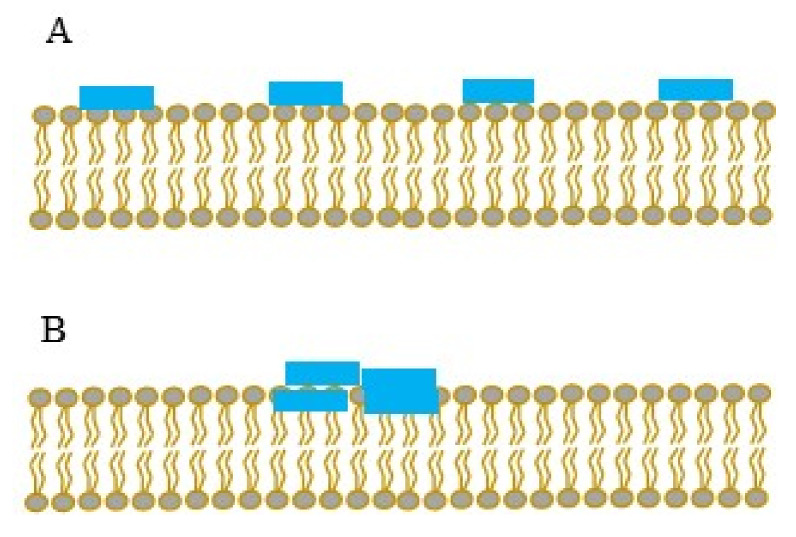
Peptide interaction with a lipid membrane with homogeneous (**A**) or heterogeneous (**B**) lateral distribution.

**Table 1 molecules-26-06442-t001:** Optical parameters obtained for a solid-supported lipid bilayer composed of an egg PC bilayer.

	*p*	*s*
Resonance minimum position shift (mdeg)	185 ± 15	83 ± 10
Thickness *t* (nm)	5.2 ± 0.1	5.2 ± 0.1
Refractive index *n*	1.52 ± 0.01	1.47 ± 0.01
Extinction coefficient *k*	0.10 ± 0.01	0.02 ± 0.002

Note: *t*, *n*, and *k* values were obtained from [12] and the other is unpublished.

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
