# Peer review of "Plasmon Waveguide Resonance: Principles, Applications and Historical Perspectives on Instrument Development"

_molecules, 2021, doi:10.3390/molecules26216442_

Round 1

Reviewer 1 Report

This review summarizes in detail the great potential of PWR in characterizing molecular interactions on sensor surfaces, including some specific applications on anisotropic thin films. This review introduces the long road and related ups and downs of PWR development from a historical perspective, which provides a good background knowledge supplement for researchers engaged in or interested in this field. In general, the relevant background knowledge in this manuscript is useful and the writing idea is clear. Therefore, I think this manuscript is worth publishing in Molecules after some minor revisions.

  1. The authorsshould examine the full text carefully to avoid misunderstanding caused by some small details. For example, line 142: “the trans-membrane helices helices”, line 403: period is missing.
  2. It is recommended to appropriately increase the theoretical description of the principle of PWR.
  3. Make more pictures to assist the understanding of some specific applications.

Reviewer 2 Report

Summary:
This manuscript reviewed the sensing mechanism, applications, and instrument development history of plasmon waveguide resonance. There are a few places of the manuscript could be further polished, such as the structure/organization and references.

Major Review:

1. Maybe the authors can give a give a short overall/broader introduction of the whole field in the beginning. 

2. The authors can also compare with other similar methods (e.g. plasmonic sensors and/or other refractive index sensors), and describe the advantage and disadvantage of the presented method.

3. The structure/organization of the manuscript can be further improved. Maybe the section of 2 and 3 should be switched with each other, e.g., history of instrument development can be moved before the applications.

4. The authors may describe a little bit more details with more figures, such as cite one example figure from each type of applications.

5. As a thorough review paper, typically much more references are expected, especially in the introduction section.

Minor Review:
1. References:
It's better to separate the sub-label letters with a ')' or '.' in the complex of references (including  [6, 8, 14-17,19. 33]), e.g. [6] a) I . D. Alves ...  

In fact, I suggest that the authors split the three references separately, otherwise it really looks weird.

Reviewer 3 Report

This review has done a comprehensive review on PWR development history and its applications. There are some comments from me:  

  1. The author might use more scientific descriptions or equations to describe the principle of PWR in the first section.
  2. The author might have a table to summarize the application in section 2.
  3. The references are not properly labeled. The author might need to reorganize them.
  4. Most of the references are from the same research group. The author might describe more PWR related works worldwide to prevent the manuscript become a personal work summary.
  5. The author might compare PWR to other similar technology in working principle, application field, and performance.

Round 2

Reviewer 2 Report

The authors have answered all my questions. I have no further comments.